# Fish Oil, Se Yeast, and Micronutrient-Enriched Nutrition as Adjuvant Treatment during Target Therapy in a Murine Model of Lung Cancer

**DOI:** 10.3390/md19050262

**Published:** 2021-05-04

**Authors:** Hang Wang, Simon Hsia, Tsung-Han Wu, Chang-Jer Wu

**Affiliations:** 1Department of Nutrition, Institute of Biomedical Nutrition, Hung-Kuang University, Taichung 433, Taiwan; 2Taiwan Nutraceutical Association, Taipei 104, Taiwan; dr.simon.hsia@gmail.com; 3Division of Hemato-Oncology, Department of Internal Medicine, Chang Gung Memorial Hospital, Keelung 204, Taiwan; u402026@gmail.com; 4Department of Food Science, Center of Excellence for the Oceans, National Taiwan Ocean University, Keelung 202, Taiwan; 5Department of Health and Nutrition Biotechnology, Asia University, Taichung 413, Taiwan; 6Graduate Institute of Medicine, Kaohsiung Medical University, Kaohsiung 807, Taiwan

**Keywords:** fish oil, Se yeast, nutraceutical formula, target therapy, adjuvant treatment, lung cancer

## Abstract

Despite the effectiveness of primary treatment modalities for cancer, the side effects of treatments, medication resistance, and the deterioration of cachexia after disease progression lead to poor prognosis. A supportive treatment modality to overcome these limitations would be considered a major breakthrough. Here, we used two different target drugs to demonstrate whether a nutraceutical formula (fish oil, Se yeast, and micronutrient-enriched nutrition; NuF) can interfere with cancer cachexia and improve drug efficacy. After Lewis lung cancer (LLC) tumor injection, the C57BL/6 mice were orally administered targeted therapy drugs Iressa and Sutent alone or combined with NuF for 27 days. Sutent administration effectively inhibited tumor size but increased the number of lung metastases in the long term. Sutent combined with NuF had no significant difference in tumor weight and metastasis compare with Sutent alone. However, NuF slightly attenuated metastases number in lung may via mesenchymal marker N-cadherin suppression. NuF otherwise increased epithelial-like marker E-cadherin expression and induce NO-mediated intrinsic apoptotic pathway in tumor cells, thereby strengthening the ability of the targeted therapy drug Iressa for inhibiting tumor progression. Our results demonstrate that NuF can promote the anticancer effect of lung cancer to targeted therapy, especially in Iressa, by inhibiting HIF-1α and epithelial-mesenchymal transition (EMT) and inducing the apoptosis of lung cancer cells. Furthermore, NuF attenuates cancer-related cachectic symptoms by inhibiting systemic oxidative stress.

## 1. Introduction

Cancer is the leading cause of death, accounting for >25% of overall death worldwide. Among the different types of cancers, lung cancer gets the first place in terms of cancer morbidity and mortality [1]. Developing an optimal therapeutic strategy for lung cancer appears to be necessary. Tyrosine kinase inhibitors (e.g., Iressa and Sutent) are regarded as targeted therapies preferentially for malignant cancers. They, however, bring on unpleasant side effects such as nausea, vomiting, hypertension, headache, etc. These side effects often influence proper functions of heart, liver, or kidney [2]. Cachexia is known the most prevalent and grave side effect in association with advanced lung cancer [3]. The occurrence of cachexia in patients with advanced lung cancer is no less than an acute malnutrition. It is estimated that ~22% of lung cancer deaths are ascribed to cachexia [4].

Supportive nutrition care for oncological patients appears to be beneficial. Some nutraceuticals exhibit chemopreventive properties, thus being conducive in view of cancer prevention. For instance, Se, an essential trace element [5], and fish oil (omega-3 fatty acid; n-3) are lauded as anti-tumorigenic nutrients equivalent to nutritional therapy [6]. Concerning a nutritional strategy for cancer cachexia patients, the diversification of nutrients is another point needed to take into account. Thus, a well-balanced nutrient formulation is critical for clinical oncology applications. In this context, we set forth to examine the therapeutic effect for a micronutrient-enriched nutrition formula (NuF) that is fortified with fish oil and Se yeast in combination with Iressa or Sutent on cancer hosts.

Tumor-promoting inflammation is a hallmark of cancer because cancer development is strongly influenced by both chronic and acute inflammation. Research of inflammation has revealed an intimate connection between inflammatory processes and neoplastic transformation, or the progression of tumor/the development of metastases and recurrences. On the other hand, medical treatments can sometimes provoke tumor cell resistance, thereby resulting in deterioration and metastases [7]. Because some nutrients and dietary components (e.g., fish oil) possess anti-inflammatory effects, whether the so-called nutritional therapy in combination with targeted therapy could exercise an added effect demands a further examination. To answer this issue, we gauged mouse inflammation cytokines and oxidative stress markers to index both local (tumor) and systemic (serum) responses when given defined nutritional supplements.

Growing evidence suggests that the epithelial-mesenchymal transition (EMT) is an important step toward tumor invasion and metastasis, and closely involved in both de novo and acquired drug resistance [8]. Cohen et al. has reported that activated human T cells could secrete such inflammatory factors as tumor necrosis factor (TNF)‑α, interleukin (IL)-6, and TGF-β, which are known able to induce EMT in inflammatory breast cancer [9]. Another important factor is hypoxia, which could cause or promote EMT. EMT is a microenvironmental condition known to promote tumor progression through stabilization of hypoxia-inducible factor-1 (HIF-1). Hence, it is interesting to know whether the combination of the targeted therapy drugs Iressa and Sutent with NuF would be able to defer or overturn the EMT process or not. In a tumor microenvironment, reactive oxygen species [10] plays an important role in cancer pathogenesis. On the contrary, chemotherapy and radiotherapy that eliminate cancer cells are considered as a result of augmented ROS generation leading cancer cells toward apoptosis [11]. Cancer cells may be more sensitive than normal cells in response to ROS accumulation. We thus hypothesize that an increased oxidative stress by exogenous ROS may selectively kill cancer cells with a relatively less damage to normal cells. Whether NuF is an antioxidant or pro-oxidant against cancer is yet another unaddressed issue.

To address this issue, we took advantage of a murine syngeneic tumor model, of which syngeneic C57BL/6 mice were subcutaneously injected with Lewis lung carcinoma (LLC) cells. We observed and recorded the progress of the experiments including tumor growth, invasion, and metastasis. Next, we evaluated the level of inflammatory and oxidative stress markers in tumor microenvironment (local). On top of that, we analyzed the concentrations of serum malondialdehyde (MDA) and NO in order to know what impact of NuF effects on systematic oxidative stress. Here, we conclude that NuF can modulate Iressa and Sutent via IL-6 and TGF-β cytokines. NuF also modulates EMT via HIF-1α. These phenomena are mutually multiplied with a net favorable effect on cellular transformation, inflammation, tumor survival, proliferation, invasion, and cancer metastasis. In a nutshell, NuF is a well-balanced nutrient formula in a position to improve targeted therapy and to ameliorate cancer cachexia.

## 2. Results

### 2.1. Cancer Cachexia

The aim of this animal experiment (Figure 1) is to understand whether a NuF (Table 1) can alleviate cancer cachexia symptoms in tumor-bearing mice treated with Iressa and Sutent (Table 2A). There were no differences in initial body weight between experimental mice. On day 28 of the experiment, the weight of tumor-bearing mice in group T drastically decreased by 11.8%; however, weight loss was alleviated in the combined groups, particularly in the TIN group for which body weight (−0.20%) was maintained after NuF intervention. The mass of gastrocnemius muscles (GM) can reflect whether there is muscular atrophy or sarcopenia. The experimental results demonstrated that among the five groups, the TIN and TSN groups have the highest GM mass (108 and 109 mg), indicating that oral administration of NuF during treatment can alleviate muscle loss in tumor-bearing mice. Cancer cachexia causes both muscle atrophy and fat loss. The results showed that there was considerable loss of white adipose tissue (WAT) in Group T while the loss of adipose tissue was slightly alleviated in tumor-bearing mice after treatment with Iressa (TI) and Sutent (TS). However, compared with Groups T and TI, the group administered with a combination of Iressa and NuF (TIN) had the most significant increase in WAT mass. In addition to WAT, we measured brown adipose tissue [12] content. The results demonstrated that BAT loss is the most severe in Group T while targeted therapy and adjuvant NuF treatment (TIN and TSN groups) resulted in a significantly higher BAT content than Group T. The increase in spleen weight, which is one of the symptoms of cancer cachexia, signifies the body is progressing to inflammation. The spleens of mice from Group T were the largest. Interestingly, targeted therapy itself cannot significantly decrease spleen weight; however, when used in combination with NuF, splenomegaly can be significantly inhibited. Thus, we reported that a combination of Iressa and NuF is the most effective in reducing body weight loss, splenomegaly, and muscle and fat loss.

### 2.2. Tumor Growth and Metastases

In terms of tumor weight, we reported that 27 days of Iressa and Sutent treatment of tumor-bearing mice (TI and TS groups) can significantly inhibit tumor size (Figure 2A). Overall, the tumor growth inhibition was the most significant in Group TIN, and its mean tumor weight was decreased by 55% compared with Group T. In terms of tumor weight distribution, the tumors in all mice from Group T weigh more than 6 g. The targeted therapy drug Sutent can inhibit tumor size but showed large variations among individual mice. Iressa treatment combined with NuF can shrunk tumors by 60% to 4 g (Figure 2B). Next, we examined tumor metastasis. First, with regard to lung metastasis, Iressa combined with NuF results in the lowest lung weight and number of metastatic foci, thus demonstrating that NuF can be used as an adjuvant treatment for Iressa to inhibit cancer cell metastasis to lungs. Interestingly, treatment of tumor-bearing mice with Sutent increased the number of lung metastases whose severity decreased when Sutent was used in combination with NuF (Figure 2C,D). With regards to liver weight, Group T has the highest liver to body ratio. However, liver weight was significantly decreased in Group TIN compared with Group T. With regards to liver metastasis, we observed metastatic foci in both Groups T and TI. No metastasis was observed after Iressa was combined with NuF. Moreover, no metastatic node number was observed in livers from the Sutent group (Figure 2F).

EMT is an important mechanism for tumor metastasis. Before metastasis, cancer cells will undergo EMT to promote invasion and migration ability in cancer cells. Therefore, we used western blot to measure the expression levels of EMT markers in tumors (Figure 3). We reported that only the combination of Iressa and NuF significantly increased epithelial-like marker E-cadherin expression and significantly decreased mesenchymal-like markers N-cadherin (Figure 3B,E). Sutent considerably reduced vimentin expression but increased the expression of slug (Figure 3C,F). These results indicate that NuF can aid in the therapeutic efficacy of Iressa and suppress tumor cells metastasis through EMT blockage. Hypoxia can promote EMT via HIF-1α. To determine the association between HIF-1α expression and EMT in tumors, we evaluated HIF-1α protein expression. As shown in Figure 3G, HIF-1α expression was diminished by combinations with Iressa and NuF (TIN) and slightly reduced when combined Sutent with NuF supplement (TSN).

### 2.3. Inflammation and Oxidation

The tumor microenvironment is a key factor for tumorigenesis and metastasis. Therefore, we measured the level of inflammatory cytokines and oxidative stress in the tumor microenvironment (Table 2B). The results showed that Iressa can decrease IL-6 concentration (although not statistically significant) compared with Group T; in combination with NuF, it can significantly decrease IL-6 and TGF-β concentrations. Thus, Sutent must be combined with NuF before IL-6 concentration is reduced. Note that there were no significant differences in TNF-α between the various groups. Within the microenvironment, oxidative stress affected tumor progression, in addition to cytokines. The study data demonstrated that the oxidation product MDA in tumors decreased after treatment but there was no significant difference. Moreover, we accidentally reported that a combination of Iressa or Sutent with NuF significantly increased NO concentration. Note that oxidative stress can be classified as systemic or tissue-specific oxidative stress. The tumor stroma can be considered to be local events while serum levels of oxidants are types of systemic oxidative stress. Systemic oxidative stress contributes to both cancer anorexia and cachexia. In terms of treatment strategy, we hope to increase local oxidative stress with simultaneous decrease in systemic oxidative stress. The results demonstrated that a combination of Iressa and NuF can significantly reduce serum MDA and NO concentrations, thus demonstrating that nutritional intervention can inhibit systemic oxidative stress.

### 2.4. Tumor Cell Apoptosis

Finally, we assessed the effects of targeted therapy drug and NuF on caspase activations in tumors (Figure 4). Caspase-3 and caspase-8 are synthesized as a 32 kDa and 45 kDa pro-form that is cleaved during activation into a large subunit of 19 and 17 kDa [13]. To determine whether caspase-3 and caspase-8 are activated in tumor, we examined for the presence of pro-form and its cleavage products using western blots. The results demonstrated that there is no significant difference in pro-caspase-3 and pro-caspase-8 levels. Iressa combined with NuF induced the tumor caspase-8 cleavage product. Although the cleaved caspase-3 does not show any significant difference, it shows an increasing trend. However, Sutent combined with NuF can result in strong active caspase-3.

## 3. Discussion

Targeted therapy is one of the most important antineoplastic strategies, and there are two types of targeted therapy drugs: epidermal growth factor receptor (EGFR), tyrosine kinase inhibitor (TKI) such as Iressa (Gefitinib) and vascular endothelial growth factor (VEGF) TKI such as Sutent (Sunitinib). The aim of this study is to assess the ability of a NuF to modulate cachexia symptom or tumor growth or progression along with different targeted drugs (Iressa or Sutent) in lung tumor-bearing mice. In order to test its capacity, we used syngeneic murine models. The LLC model’s advantage is that, unlike the widely used xenograft models in which human cells are implanted into the mouse tissue, implanted cells are immunologically compatible with the murine system. Consequently, LLC models can be created on an immunocompetent murine background, such as C57BL, and true immune and toxicity responses can be evaluated with respect to targeted therapies and tumor growth. Moreover, because the LLC model can be syngeneic, the tumor microenvironment can be accurately depicted in the animal model.

Cachexia is a complex metabolic syndrome related to the underlying disease, which is characterized by significantly decreased body weight and depleted muscle mass and adipose tissue over a short time period. Systemic inflammation is commonly observed in patients with cancer cachexia, as it has been postulated to play an important role in the etiology of the condition and in the determination of clinical symptoms [14]. In general, the nutritional status is negatively correlated to the severity of cancer cachexia, as both are clinically relevant in projection to an overall therapeutic outcome. If side effects aggravated, it would neither warrant patient’s living quality nor a positive prognosis. Pappalardo et al. (2015) reviewed the EPA issue as an anti-inflammatory agent, thus concluding that EPA supplementation has a positive effect in stabilizing lean body mass compared to standard supplementation by diminishing C-reactive protein, IL-6, and TNF levels [15]. NuF is a complete nutrition formula that contains multi-nutraceuticals such as whey protein, coenzyme Q10, fish oil, and selenium yeast (Table 1). Our previous studies discovered that by feeding, the complete nutrient formula can effectively alleviate cachexia induced by tumor [16] or chemotherapy [17] in xenograft models. In addition, it is known that omega-3 fatty acids can improve cachexia in humans including in humans with advanced lung cancer who have undergone chemotherapy [18,19]. However, how NuF works with TKI is still unclear. Therefore, this study sought to test the hypothesis that NuF would produce additional benefits compared to the conventional target therapy alone. Here we demonstrate that Iressa or Sutent combined with NuF increase GM, WAT, and BAT results in an improvement in weight loss and cachexia (Table 2A).

There are multiple sources of inflammatory factors, including the tumor and the various peripheral organs, tissues, and cells. Local tumor tissue inflammation may induce the expression of proinflammatory cytokines, which can further promote tumor progression. This tumor microenvironment produces various factors such as interleukin-6 (IL-6), TNF, and TGF, all of which stimulate a transient EMT to promote cancer progression, invasion, and metastasis [20]. EMT is a biological process in which a non-motile epithelial cell changes to a mesenchymal phenotype with invasive capacities. This phenomenon has been documented in multiple biological processes, including tumor progression and metastasis. The hallmark of EMT is the loss of epithelial surface markers, most notably E-cadherin, and the acquisition of mesenchymal markers such as vimentin and N-cadherin. The downregulation of E-cadherin during EMT can be mediated by its transcriptional repression by the binding of EMT transcription factors such as snail and slug to E-boxes present in the E-cadherin promoter [21]. A hypoxic environment is present in different types of tumors, especially in solid tumors. To adjust to the hypoxic microenvironment, several cancer cells increase the production of hypoxia-inducible factors (HIFs), which are associated with increased malignancy, poor prognosis, and resistance to radiotherapy and chemotherapy [22]. The study demonstrate that HIF-1 is implicated in the regulation of several genes involved in the multiple key steps of metastasis, including EMT, invasion, extravasation, and metastatic niche formation, mostly in solid tumors [23]. Therefore, we tested the effect of NuF on HIF-1 protein expressions in LLC tumors.

We reported that TIN mice showed a significantly decrease in IL-6 and TGF-β, increase in the expression of epithelial markers (E-cadherin), and decrease in the expression of mesenchymal markers (N-cadherin) in tumor microenvironment (Table 2 and Figure 3). A master molecule in the EMT induction appears to be TGF-β, however, the number of molecules and routes implicated in EMT is still growing [24]. Recently, studies reported that vitamin D attenuated TGF-β-induced pro-fibrotic effects by inhibiting EMT in human alveolar epithelia A549 cells [25]. Another study reported the co-administration of coenzyme Q10 and sitagliptin suppression of TNF-α, TGF-β in renal tissue on experimentally induced diabetic nephropathy in rats [26]. We identified that the TIN group showed a ~60% HIF-1 expression compared with T group (Figure 3G). Moreover, the relevant experiments display considerable synergistic efficiency in the co-administration of NuF and Iressa with increase in epithelial markers (E-cadherin) through TGF-β and IL-6 inhibition. In addition, there is emerging evidence that omega-3 fatty acids may play an integral role in linking the microbiome and immune system by regulating inflammation [27]. A recent study finds a link between microbiota and response to Sutent [28] as well as Iressa [29]. Therefore, NuF may increase the efficacy of TKIs by regulating the gut microbiota. but further research is needed to confirm.

A study by Conley et al. demonstrated that the administration of antiangiogenic agents such as the VEGF receptor TKI sunitinib and the anti-VEGF antibody bevacizumab increases the cancer stem cells population in breast cancer xenografts because of the generation of tumor hypoxia [30]. Similarly, when Sutent was tested, although Sutent causes inhibition of tumor growth and mesenchymal markers (e.g., vimentin), the expression of EMT-regulating transcription factor slug was induced (Figure 3C,F). Nevertheless, along with NuF (TSN group), Sutent showed attenuated responses of reducing IL-6, N-cadherin, vimentin, and HIF1 expression (Figure 3). A recent study reported that IL-6 can induce EMT of non-small cell lung cancer cells by STAT3 activation and insulin-like growth factor-1 receptor (IGR-1R) [31]. Since DHA destabilizes HIF-1, its proteolytic degradation is promoted via PPAR activation [32]. Therefore, the administration of NuF may enhance Sutent efficiency by targeting HIF-1α signaling and IL-6-mediated EMT.

Note that the reactive species or free radicals include reactive oxygen and nitrogen species, which are collectively termed as reactive oxygen nitrogen species. During tumor progression, oxidative stress increases, leading to secondary effects that compromise the patient’s quality of life. Oxidative stress-induced lipid peroxidation generates numerous electrophilic aldehydes (such as MDA) that can attack many cellular targets. These products of oxidative stress can decrease cell cycle progression of cancer cells and lead to cell cycle checkpoint arrest; moreover, the effects may interfere with the ability of anticancer drugs to kill cancer cells [33]. Note that supplementation with antioxidants as prophylactic agents can help in cancer prevention and treatment [34]. Several phenolic compounds may exert both effects, i.e., chemopreventive and anticancer, because of their dual effect on cellular redox regulation. It seems that the phenolic compounds promote an antioxidant effect, which leads to the prevention of carcinogenesis in normal cells, and they exert a pro-oxidant effect that favors cell death in cancer cells [35]. Therefore, we examined whether targeted therapy drug combined with NuF has a pro-oxidant action on tumors, leading to cell death, or possesses antioxidant capacity to alleviate cachexia. The results showed that a combination of drugs and NuF can significantly decrease the concentration of serum oxidation products MDA and NO. The phenomenon is most significant in the TIS group. However, a combination of drug and NuF has no effect on tumor MDA levels but significantly increases NO (Table 2B).

Reactive nitrogen species are generated from physiological processes to produce metabolites and energy as a defense against invasive microorganisms [36]. Moreover, a series of studies demonstrated that endogenously produced or exogenously supplied NO activates caspases along with apoptosis in various tumor cells [37,38]. Ovadje et al. demonstrated that NO-induced caspase-8 activation in human leukemia cells and caspase-3, caspase-6, and caspase-9 activation in human breast cancer cells [39]. Moreover, the NO was used as a target to sensitize tumor cells; we observed a similar phenomenon, i.e., NuF synergistic effect with targeted drug on NO induction. NuFu synergizes with Iressa to induce caspase-8 activation, and along with Sutent can significantly activate caspase-3 (Figure 4). The data show that NuF combined with Iressa or Sutent can stimulate NO-induced ROS generation, thus leading to caspase deglutathionylation and activation, subsequently activating tumor cell death.

We demonstrate that NuF has different adjuvant effect on target therapy. NuF combined with Iressa demonstrations more effective than Sutent. NuF can inhibiting IL-6 and TGF-β trigger EMT that influences the HIF-1α activity for inhibiting cancer progression during Iressa therapy. Moreover, NuF may both mediate tumor cell apoptotic death and protect systemic oxidative stress in tumor-bearing mice undergoing therapy. Our results suggest that NuF is a promising candidate for a nutrition formula that targets HIF-1 signaling and causes apoptosis to improve the Iressa treatment for lung cancers.

## 4. Materials and Methods

### 4.1. Cell Line and Animals

LLC cells were purchased from Bioresource Collection and Research Center (Hsinchu, Taiwan). Cells were cultured in DMEM (Sigma-Aldrich) supplemented with 10% FBS (Sigma-Aldrich), 2 mmol/L l-glutamine, 100 U/mL penicillin, and 100 mg/mL streptomycin. Male C57BL/6 mice (6–7 weeks) were obtained from the National Laboratory Animal Center (Taipei, Taiwan). Mice were individually housed in a climate-controlled room (12:12 dark-light cycle with a constant room temperature of 21 ± 1 °C). They underwent at least four-day adjustment to a new environment and diet before the treatments were performed. Mice were allowed free access to water and food (Laboratory Rodent Diet, LabDiet, St. Louis, MO, USA).

### 4.2. Experimental Diet

A NuF that contains several ingredients, including fish oil (omega-3 fatty acid), selenium, glutamine, and CoQ1, was obtained from New Health Enterprise Inc. (Tustin, CA, USA). Its contents are listed in Table 1.

### 4.3. Experimental Design

After acclimatization, mice were divided into weight-matched groups and inoculated subcutaneously (s.c.) with the suspension of LLC tumor cells (3 × 10^5^) on day 7. After day 7, mice orally administered for 27 days and distributed into five groups (Figure 1): the no treatment (T) group, TI group (Iressa 30 mg/kg/day), TIN group (Iressa 30 mg/kg/day and NuF 1 g/mouse/day), TS group (Sutent 10 mg/kg/day), and TSN group (Sutent 10 mg/kg/day and NuF 1 g/mouse/day). After inoculating tumor cells, body mass, food intake, and tumor size were measured three times a week.

Tumor growth was then assessed using a caliper and the inner diameter was recorded between the two rulers for each tumor. The tumor volume was calculated using the following equation: tumor volume (mm^3^) = width × length^2^/2. The animals were sacrificed by the CO2 inhalation method on day 28. Lungs were then harvested after euthanasia and the surface tumors were counted. Organs were removed, weighed, and stored at −20 °C for additional analysis. Murine blood was collected; centrifuged at 4 °C, 3500 rpm for 30 min; and then the suspension was collected to measure the serum levels of MDA and NO.

The experimental procedures were approved by the Animal Ethical Committee: Institutional Animal Care and Use Committee (IACUC) of the Hung-Kuang University of College of Medical and Health Care, Approval Number: HK105-101 and followed the principles of good laboratory animal care. The animal experiments complied with the guidelines for maintaining and handling of experimental animals established by the Institutional Animal Care and Use Committee (IACUC) of the Hung-Kuang University of College of Medical and Health Care (HK105-101). The carcass weight was calculated by subtracting the tumor weight from body weight, and final body weight gain was calculated from the difference between carcass weight and initial weight.

### 4.4. Measurement of Oxidative Stress

MDA products, the metabolites of polyunsaturated fatty acids, were assessed as a marker for lipid peroxidation. Briefly, the serum or supernatant from tissue homogenates was mixed with 3% SDS, 0.1 N HCl, 10% phosphotungstic acid, and 0.7% thiobarbituric acid, and then incubated at 95 °C for 60 min as per a previously reported method [40]. n-butanol was then added, and the mixture was vigorously shaken. After centrifugation, thiobarbituric acid-reactive substances in the n-butanol layer were taken and measured with Wallac 1420 multilabel counter Victor 3 (PerkinElmer, Turku, Finland) at 530 nm with excitation at 485 nm. MDA levels were calculated using 1,1,3,3-tetraethoxypropane as a standard.

The nitric oxide assay was performed as previously described with slight modifications [10]. The amounts of nitrite, a stable metabolite of NO, were measured using Griess reagent (1% sulphanilamide and 0.1% naphthylethylenediamine dihydrochloride in 2.5% phosphoric acid). Briefly, 50 µL of serum or supernatant from tissue homogenates were mixed with 50 µL of Griess reagent. Subsequently, the mixture was incubated at room temperature for 10 min and the absorbance at 550 nm was measured in a microplate reader.

### 4.5. Measurement of Cytokines in Tumor Tissue

Tumor tissue samples from each mouse were sonicated at 4 °C in 500–2000 µL of 20 mM Tris–HCl, pH 7.5, 2 mM ATP, 5 mM MgCl_2_, and 1 mM dithiothreitol (DTT), and 5 µL of a protease inhibitor cocktail (Sigma)/mL of buffer. They were then centrifuged at 10,000× *g* for 30 min, and the supernatants were collected for further analysis. A total of three cytokines, namely, IL‑6, TNF‑α, and TGF-β, from tumor tissue homogenates were measured using a quantitative sandwich enzyme technique with Quantikine^®^ ELISA kits (BD Biosciences), as per the manufacturer’s protocol.

### 4.6. Western Blot

Western blot was used to observe the expression of E-cadherin, N-cadherin, β-catenin, vimentin, snail, slug, caspase-3, caspase-8, and caspase-3 cleaved in a tumor microenvironment. The inoculated tumor was resected and treated with a lysis buffer solution (20 mM Tris–HCl, pH 7.5, 2 mM ATP, 5 mM MgCl_2_, 1 mM dithiothreitol) and protease inhibitor cocktail (Sigma-Aldrich, Saint Louis, MI, USA) to obtain tumor proteins. Sodium dodecyl sulfate-polyacrylamide gel electrophoresis (SDS-PAGE) was applied for western blots. Murine anti-E-Cadherin (Cell Signaling, Danvers, MA, USA), anti-N-cadherin (Cell Signaling, MA, USA), anti-Vimentin (Cell Signaling, MA, USA), anti-Slug (Cell Signaling, MA, USA), anti- Snail (Cell Signaling, MA, USA), anti-Caspase-3 (Cell Signaling, MA, USA), anti-Caspase-9 (Cell Signaling, MA, USA), anti-Caspase-3 cleaved (Cell Signaling, MA, USA), anti-Caspase-9 cleaved (Cell Signaling, MA, USA), anti-HIF-1α (Cell Signaling, MA, USA), and anti-β-Actin (Cell Signaling, MA, USA) antibodies were used to label tumor proteins. Moreover, the tumor proteins were treated with the Pierce BCA protein assay kits (Thermo Fischer, Waltham, MA, USA), and their absorbance were measured at 550 nm.

### 4.7. Statistical Analysis

Data were expressed as means ± standard deviation. Statistical significance was determined by one-way ANOVA using Bonferroni’s multiple comparison test (Prism Graph Pad). Differences were considered to be statistically significant when *p* < 0.05.

## 5. Conclusions

To summarize, we used an immunocompetent lung carcinoma mouse model to examine whether omega-3 fatty acid, se yeast, and micronutrient-enriched nutrition supplement can strengthen the efficacy of targeted therapy. The results showed that the special nutrient supplement can alleviate cachexia in cancer and improve the efficacy of targeted therapy. In view of this, this special nutrient supplement can be an option for adjuvant therapy in cancer patients.

## Figures and Tables

**Figure 1 marinedrugs-19-00262-f001:**
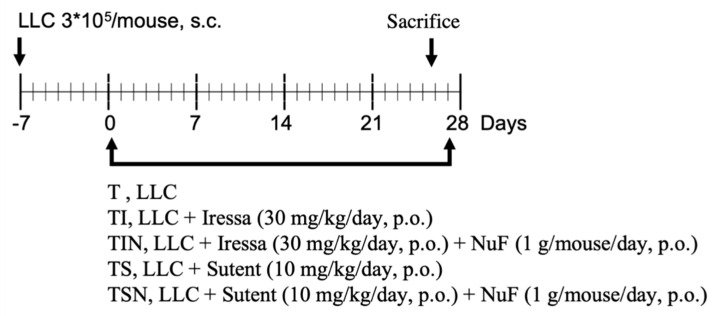
The treatment protocol of the nutraceutical formula (NuF) combined with targeted therapy in tumor-bearing mice. Lewis lung carcinoma (LLC): 3 × 105 cells were subcutaneously into the right side of the back of C57BL/6 mice on day 7. Tumor volume was calculated using the following formula: 1/2 (*xy^2^*), wherein *x* = tumor width and *y* = tumor length. When tumor size approached 50 mm^3^, it was considered as day 0. The tumor-bearing mice were randomized into the following five groups: the no treatment (T) group, TI group (Iressa 30 mg/kg/day, orally administered for 27 days), TIN (Iressa 30 mg/kg/day and NuF 1 g/mouse/day, orally administered for 27 days), TS (Sutent 10 mg/kg/day, orally administered for 27 days), and TSN (Sutent 10 mg/kg/day and NuF 1 g/mouse/day, orally administered for 27 days). After 28 days, all mice were sacrificed, the inoculated tumors, gastrocnemius muscle, white adipose tissue, brown adipose tissue, liver, lung, and spleen were then resected for further examination.

**Figure 2 marinedrugs-19-00262-f002:**
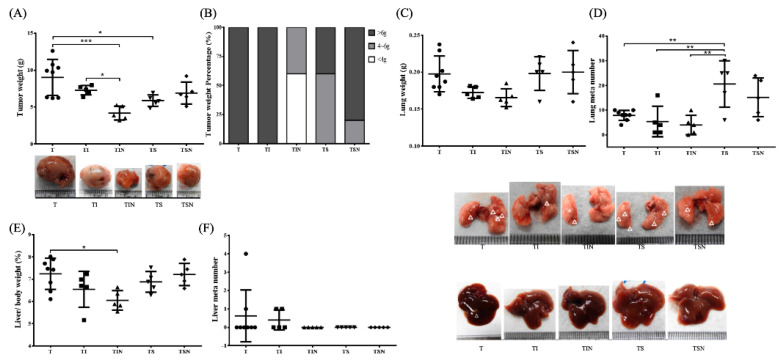
Anticancer and anti-metastatic effect of the combination of tyrosine kinase inhibitors (TKIs) and NuF in tumor-bearing mice. (**A**) tumor weight. (**B**) tumor weight distribution. (**C**) lung weight. (**D**) The average numbers of lung metastatic nodules. (**E**) Liver weight as a percentage of the whole body weight. (**F**) The average numbers of liver metastatic nodules. Representative photos of the lungs and livers. The arrows point to the metastatic nodules. T, tumor no treatment; TI, tumor treated with Iressa; TIN, tumor treated with Iressa combined with NuF; TS, tumor treated with Sutent; TSN, tumor treated with Sutent combined with NuF. * *p* < 0.05, ** *p* < 0.01, and *** *p* < 0.001 denote levels of significant differences between groups. Data are expressed as means ± SD.

**Figure 3 marinedrugs-19-00262-f003:**
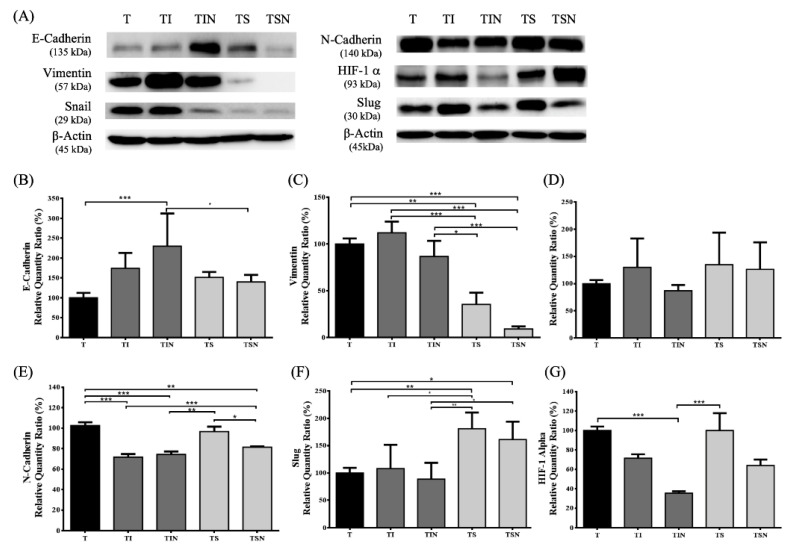
Co-administration of NuF and Iressa inhibited the epithelial-mesenchymal transition (EMT) of tumor in vivo. (**A**) The graph represents relative densitometric intensity of each band normalized to β-actin. (**B**–**F**) Quantitative analysis of the EMT markers: E-cadherin, Vimentin, Snail, and Slug in tumor was verified by western blot analysis. (**G**) Quantitative analysis of the HIF-1α in tumor was verified by western blot analysis. Data are shown as mean ± SD. *n* = 5–8 mice/group and each value is an average of three independent experiments. * *p* < 0.05, ** *p* < 0.01, and *** *p* < 0.001 denote levels of significant differences between groups.

**Figure 4 marinedrugs-19-00262-f004:**
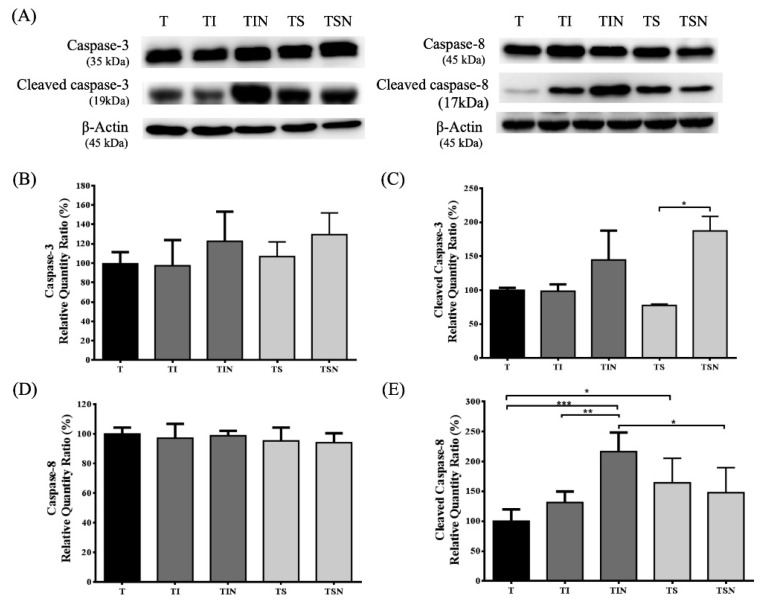
Co-administration of NuF and TKIs induced the apoptosis of tumor in vivo. (**A**) Western blot analyses of caspase-3/cleaved caspase-3, caspase-8/cleaved caspase-8. (**B**)–(**E**) Quantitative analysis of the pro-caspase-3, caspase-3 p19, pro-caspase-8, caspase-8 p17 in tumor were verified by western blot analysis. Data are shown as mean ± SD. *n* = 5–8 mice/group and each value is an average of three independent experiments. * *p* < 0.05, ** *p* < 0.01, and *** *p* < 0.001 denote levels of significant differences between groups.

**Table 1 marinedrugs-19-00262-t001:** Components of nutraceutical formula (NuF).

Nutrawell powder	1 Serving: 75 g
Components (Units)	Amount per Serving
Calorie (kcal)	294
Protein (g)	19
Fat (g)	8.7
Saturated fatty acids (g)	1.1
Monounsaturated fatty acids (g)	3.7
Polyunsaturated fatty acids (g)	3.9
Fish oil (Omega-3 fatty acid; n-3) (g)	1.4
Trans fatty acids (g)	0
Carbohydrates (g)	37
Diet fiber (g)	4.2
Sugar (g)	3
Sodium (mg)	360
Cholesterol (mg)	0
Vitamin A (µg)	350 (1167 IU)
β-Carotene (µg)	300 (1000 IU)
Vitamin B_1_ (mg)	1.4
Vitamin B_2_ (mg)	1.6
Vitamin B_6_ (mg)	1.5
Vitamin B_12_ (µg)	2.4
Vitamin C (mg)	200
Vitamin D (µg)	5 (200 IU)
Vitamin E (mg)	13, d-α TE (19 IU)
Vitamin K (µg)	37.5
Niacin (mg)	13.5
Folic acid (µg)	600
Pantothenic acid (mg)	5
Biotin (µg)	100
Choline (mg)	300
Coenzyme Q_10_ (mg)	30
Calcium (mg)	480
Phosphorous (mg)	250
Iron (mg)	4.5
Iodine (µg)	52
Magnesium (mg)	117
Zinc (mg)	7.5
Selenium yeast (µg)	110
Potassium (mg)	682
Copper (µg)	400
Manganese (µg)	1.5
Chromium (µg)	90
Molybdenum (µg)	56.3

**Table 2 marinedrugs-19-00262-t002:** Cachectic, immune and oxidative parameters between groups.

**(A) Physiological Cachectic Parameters**								
**Treatment**	***N***	**IW (g)**	**CW (g)**	**Dela CW (%)**	**GM (mg)**	**WAT (mg)**	**BAT (mg)**	**Spleen (mg)**
T	8	21.3 ± 1.3	18.80 ± 1.58	−11.80 ± 9.0	86 ± 11	13 ± 17	20 ± 14	480 ± 121
TI	5	20.3 ± 1.4	18.28 ± 1.56	−8.20 ± 10.1	88 ± 10	36 ± 21	43 ± 11	354 ± 35
TIN	5	20.9 ± 1.5	20.86 ± 2.06	−0.20 ± 9.8 *	108 ± 7	72 ± 29 **^,+^	57 ± 9 *	254 ± 50 **
TS	5	21.3 ± 1.2	18.17 ± 1.49	−7.90 ± 7.8	86 ± 15	39 ± 18	55 ± 10*	403 ± 78
TSN	5	21.9 ± 1.5	19.15 ± 1.00	−4.35 ± 6.6	109 ± 38	64 ± 21 **	63 ± 31 **	260 ± 66 **
**(B) Immune and Oxidative Parameters**								
**Treatment**	***N***	**Tumor**					**Serum**	
		**IL-6 (pg/mL)**	**TNF-α (pg/mL)**	**TGF-β (pg/mL)**	**MDA (uM)**	**NO (%)**	**MDA (μM)**	**NO (%)**
T	8	5.3 ± 1.7	51.6 ± 18.2	35.9 ± 5.9	51.1 ± 28.6	100 ± 28.6	6.0 ± 4.5	100 ± 19.7
TI	5	4.7 ± 2.9	47.8 ± 9.9	34.6 ± 3.8	49.3 ± 28.6	111.4 ± 30.7	4.2 ± 1.0 *	78.1 ± 15.2 **
TIN	5	3.2 ± 1.1 ^#^	45.9 ± 11.4	28.6 ± 6.7 *	40.3 ± 26.0	128.4±24.6 *^,+^	2.3 ± 0.8 *	65.8 ± 20.2 **
TS	5	6.5 ± 2.5	49.2 ± 11.8	34.6 ± 3.2	47.2 ± 30.5	99.9 ± 34.5	8.5 ± 5.1	89.4 ± 18.9
TSN	5	3.0 ± 0.8 ^#^	44.7 ± 22.2	30.0 ± 4.3	48.9 ± 35.4	121.2 ± 26.2 *^,#^	7.1 ± 2.6	78.6 ± 13.4 *

IW = Initial body weght, CW = Carcass weight, Dela CW = Final body weight gain (%), GM = Gastrocnemius, WAT = White adipose tissue, BAT = Brown adipose tissue, MDA = Malondialdehyde. Data represent mean ± SD. * *p* < 0.05, ** *p* < 0.01 presents significant differences from T. ^+^ *p* < 0.05 presents significant differences from TI. ^#^ *p* < 0.05 presents significant differences from Ts. Statistical significance was determined by one-way ANOVA.

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
