# Peer review of "Fish Oil, Se Yeast, and Micronutrient-Enriched Nutrition as Adjuvant Treatment during Target Therapy in a Murine Model of Lung Cancer"

_marinedrugs, 2021, doi:10.3390/md19050262_

Round 1

Reviewer 1 Report

The manuscript “Fish oil, Se yeast and micronutrient-enriched nutrition as adjuvant treatment during target therapy in a murin model of lung cancer” is suitable form publication to the journal Marine Drugs.  The authors described a systematic and rational approach in demonstrating a nutrient mixture, NuF as a potential adjuvant treatment for cancer therapy in combination with known anti-cancer drugs, Iressa and Sutent, in a murine animal model.

However, the authors overlooked a certain area that needs worth mentioning in their discussion.  One of the major developments in immunotherapy is the role of the microbiome in the immune modulation of cancer and the role of inflammation.  Although the manuscript does not focus on the effect of the nutrient mixture, NuF, on the immune response but mentions inflammation, it is inevitable not to consider these in a broader context, since one of NuF's major components is fish oil or omega-3 fatty acids.   Moreover, the authors cited their use of the selected animal model as one of the advantages of using syngeneic mice to reveal any immune responses.

Iressa or gefitinib and its metabolites has been shown to affect the microbiome (Wang C et al https://doi.org/10.1371/journal.pone.0236523) as well as Sutent or sunitinib (Gong J et al. DOI: 10.1200/JCO.2018.36.6_suppl.657 Journal of Clinical Oncology 36, no. 6_suppl (February 20, 2018) 657-657.)

There is emerging evidence that omega-3 fatty acids may play an integral role in linking the microbiome and immune system by regulating inflammation (Ilag, L.L. Are Long-Chain Polyunsaturated Fatty Acids the Link between the Immune System and the Microbiome towards Modulating Cancer? Medicines 2018, 5, 102. https://doi.org/10.3390/medicines5030102).  This extends from the results of the authors.

In addition, it is known that omega-3 fatty acids can improve cachexia in humans including in humans with advanced lung cancer who have undergone chemotherapy. Thus, the authors should include citations of the relevant references such as :

  1. Smith GI, Julliand S, Reeds DN, Sinacore DR, Klein S, Mittendorfer B. Fish oil-derived n-3 PUFA therapy increases muscle mass and function in healthy older adults. Am J Clin Nutr. 2015 Jul;102(1):115-22. doi: 10.3945/ajcn.114.105833. Epub 2015 May 20. PMID: 25994567; PMCID: PMC4480667.
  2. Finocchiaro C, Segre O, Fadda M, Monge T, Scigliano M, Schena M, Tinivella M, Tiozzo E, Catalano MG, Pugliese M, Fortunati N, Aragno M, Muzio G, Maggiora M, Oraldi M, Canuto RA. Effect of n-3 fatty acids on patients with advanced lung cancer: a double-blind, placebo-controlled study. Br J Nutr. 2012 Jul;108(2):327-33. doi: 10.1017/S0007114511005551. Epub 2011 Nov 25. PMID: 22114792.

Although it is not necessary to conduct additional experiments to demonstrate the effect of the treatments on the microbiome, proper citation of the references mentioned above in the manuscript’s Discussion section should be sufficient prior to acceptance for publication.

Author Response

We thank the reviewer’s suggestion. We have included reviewer' suggestions in the discussion of the article.

Reviewer 2 Report

COMMENTS MANUSCRIPT NUMBER 1183772 :

In this paper the authors used two different target drugs (Iressa and Sutent) to demonstrate whether a nutraceutical formula (fish oil, Se yeast, and micronutrient-enriched nutrition; NuF) can interfere with cancer cachexia and improve drug efficacy. Sutent administration effectively inhibited tumor size but increased the number of lung metastases in the long term. The combination of NuF and Iressa significantly inhibited HIF-1α and IL-6 expression and induced tumor NO and apoptosis, resulting

reduced tumor and metastases.  I consider this study to be very interesting. However, many issues have to be addressed.   The introduction, results, and discussion section are very poor, and the interpretations and conclusions are not appropriate because they are not consistent with the results that the authors show. 

Comment 1. The abstract is unclear and it is not consistent with the results. 

Comment 2. The introduction section is very long and most of the information that the authors present in this section can be in the discussion.  

Comment 3. In the results section, the author mentioned the measurement of mice weight and the GM in the different experimental groups, however, they did not show the results.  

Comment 4. How can the authors get LLC-derived tumors with 6 g? It has been reported that in LLC derived tumors with 3000-4000 mm3 (that is so big) size, the weight is not more than 4 g and at that size of the tumor, the mice are in very unhealthy condition.

Comment 5. In the figure legend of figure 2, the authors have to present what treatment corresponded to T, TI, TIN, etc.

Comment 6.  In figure 2, the liver weight cannot be used as a measure of metastasis.

Comment 7. In figure 4, the authors mentioned caspase-9, but they did not present the results. I considered that these results are important to be presented to have a better picture of the two pathways of the apoptosis.

Comment 8.  The Table 2A has to be referenced in the results section.

Comment 9: Why do the authors present the results using Sutent+NuF if that result has no significant difference?

Comment 10: The authors have to clarify the number of LLC cells that they used in the mice model. In the scheme model they mentioned 3X105 and in material and methods the mentioned 1X105. In this model, it is very important to know the tumor cells number injected.

Comment 11: The authors mentioned the combination of Sutent+NuF significantly inhibited HIF-1α and IL-6 expression and induced tumor NO and apoptosis, that induced reduction of tumor and metastases. However, the results that the authors show demonstrated, that the combination of Iresa and NuF has this effect and not the Sutent+NuF combination. The authors have to clarify this point.

Author Response

Response to Reviewer 2 Comments

Comment 1. The abstract is unclear and it is not consistent with the results.

Response 1: We thank the reviewer’s suggestion. The abstract has been carefully corrected.

Comment 2. The introduction section is very long and most of the information that the authors present in this section can be in the discussion. 

Response 2: We thank the reviewer’s suggestion. The introduction has been streamlined.

Comment 3. In the results section, the author mentioned the measurement of mice weight and the GM in the different experimental groups, however, they did not show the results. 

Response 3: We thank the reviewer’s reminder. Those results are placed in Table 2. We insert the ‘Table 2A’ in the result section.

Comment 4. How can the authors get LLC-derived tumors with 6 g? It has been reported that in LLC derived tumors with 3000-4000 mm3 (that is so big) size, the weight is not more than 4 g and at that size of the tumor, the mice are in very unhealthy condition.

Response 4: We thank the reviewer’s suggestion. Because the mice showed normal motility and no deaths were observed. We would like to thank the reviewer for your valuable suggestion and will pay more attention to the condition of the mice in future experiments.

Comment 5. In the figure legend of figure 2, the authors have to present what treatment corresponded to T, TI, TIN, etc.

Response 5: We thank the reviewer’s suggestion. Treatment for each group has been present in the figure legend of Figure 2.

Comment 6. In figure 2, the liver weight cannot be used as a measure of metastasis.

Response 6: We thank the reviewer’s suggestion. Liver weight only as a reference data of liver pathological status. We mainly use numbers of liver metastatic nodules to predict metastatic severity.

Comment 7. In figure 4, the authors mentioned caspase-9, but they did not present the results. I considered that these results are important to be presented to have a better picture of the two pathways of the apoptosis.

Response 7: We thank the reviewer’s suggestion. The study demonstrated that NO-induced caspase-8 [1], since we find NuF can induced NO expression, thus we examine whether NuF can induce the activation of caspase-8.

Reference

[1] Ovadje, P.; Chatterjee, S.; Griffin, C.; Tran, C.; Hamm, C.; Pandey, S., Selective induction of apoptosis through activation of caspase-8 in human leukemia cells (Jurkat) by dandelion root extract. J Ethnopharmacol 2011, 133, (1), 86-91.

Comment 8.  The Table 2A has to be referenced in the results section.

Response 8: We thank the reviewer’s suggestion. The corresponding information has been included in 2.1 Cancer cachexia.

Comment 9: Why do the authors present the results using Sutent+NuF if that result has no significant difference?

Response 9: We thank the reviewer’s suggestion. We have revised results more precisely in the article.

Comment 10: The authors have to clarify the number of LLC cells that they used in the mice model. In the scheme model they mentioned 3X105 and in material and methods the mentioned 1X105. In this model, it is very important to know the tumor cells number injected.

Response 10: We thank the reviewer’s suggestion. Once I typed 1X105, when I really meant 3X105. Sorry for the spelling error. The corresponding information has been included in 4.3 Experimental design.

Comment 11: The authors mentioned the combination of Sutent+NuF significantly inhibited HIF-1α and IL-6 expression and induced tumor NO and apoptosis, that induced reduction of tumor and metastases. However, the results that the authors show demonstrated, that the combination of Iresa and NuF has this effect and not the Sutent+NuF combination. The authors have to clarify this point.

Response 11: We thank the reviewer’s suggestion. We have recorrected the results and the corresponding information has been included in the section of Abstract and Discussion. 

Round 2

Reviewer 2 Report

COMMENTS OF SECOND REVISION OF THE MANUSCRIPT NUMBER 1183772 :

Comment 1. The authors answered all the comments. However, I did not see the how they streamlined the introduction.  

Author Response

Comment 1. The authors answered all the comments. However, I did not see the how they streamlined the introduction.

Response 1: We thank the reviewer’s suggestion.

  • In the first revised version we moved
  1. Targeted therapy is one of the most important antineoplastic strategies, and there are two types of targeted therapy drugs: epidermal growth factor receptor (EGFR) tyrosine kinase inhibitor (TKI) such as Iressa (Gefitinib) and vascular endothelial growth factor (VEGF) TKI such as Sutent (Sunitinib).
  2. Cachexia is a complex metabolic syndrome related to the underlying disease, which is characterized by significantly decreased body weight and depleted muscle mass and adipose tissue over a short time period. Systemic inflammation is commonly observed in patients with cancer cachexia, as it has been postulated to play an important role in the etiology of the condition and in the determination of clinical symptoms.
  3. The hallmark of EMT is the loss of epithelial surface markers, most notably E-cadherin, and the acquisition of mesenchymal markers such as vimentin and N-cadherin. The downregulation of E-cadherin during EMT can be mediated by its transcriptional repression by the binding of EMT transcription factors such as snail and slug to E-boxes present in the E-cadherin promoter.

from introduction to discussion. We have also tried to streamline the introduction. Word count dropped from 1002 to 762.

  • In this latest version we moved
  • In general, the nutritional status is negatively correlated to the severity of cancer cachexia, as both are clinically relevant in projection to an overall therapeutic outcome. If side effects aggravated, it would neither warrant patient’s living quality nor a positive prognosis.

from introduction to conclusion. We also reedit the content of the introduction more accurately, and word count dropped from 762 to 731. We hope this revision can meet reviewer standards. 
